# Select neurotrophins promote oligodendrocyte progenitor cell process outgrowth in the presence of chondroitin sulfate proteoglycans

Justin R. Siebert[1]     |    Donna J. Osterhout[2]

[1]Physician Assistant Program, Department of Biology, Slippery Rock University, Slippery Rock Pennsylvania, Slippery Rock, PA, USA

[2]Department of Cell and Developmental Biology, SUNY Upstate Medical University, Syracuse, NY, USA

**Correspondence**
Justin R. Siebert, Physician Assistant Program, Department of Biology, Slippery Rock University, Pennsylvania, 121 Harrisville Building, 1 Morrow Way, Slippery Rock, PA 16057, USA.
Email: justin.siebert@sru.edu

**Funding information**
New York State Department of Health; Craig H. Neilsen Foundation

## Abstract

Axonal damage and the subsequent interruption of intact neuronal pathways in the spinal cord are largely responsible for the loss of motor function after injury. Further exacerbating this loss is the demyelination of neighboring uninjured axons. The post-injury environment is hostile to repair, with inflammation, a high expression of chondroitin sulfate proteoglycans (CSPGs) around the glial scar, and myelin breakdown. Numerous studies have demonstrated that treatment with the enzyme chondroitinase ABC (cABC) creates a permissive environment around a spinal lesion that permits axonal regeneration. Neurotrophic factors like brain-derived neurotrophic factor (BDNF), glial cell line-derived neurotrophic factor (GDNF), neurotrophic factor-3 (NT-3), and ciliary neurotrophic factor (CNTF) have been used to promote neuronal survival and stimulate axonal growth. CSPGs expressed near a lesion also inhibit migration and differentiation of endogenous oligodendrocyte progenitor cells (OPCs) in the spinal cord, and cABC treatment can neutralize this inhibition. This study examined the neurotrophins commonly used to stimulate axonal regeneration after injury and their potential effects on OPCs cultured in the presence of CSPGs. The results reveal differential effects on OPCs, with BDNF and GDNF promoting process outgrowth and NT-3 stimulating differentiation of OPCs, while CNTF appears to have no observable effect. This finding suggests that certain neurotrophic agents commonly utilized to stimulate axonal regeneration after a spinal injury may also have a beneficial effect on the endogenous oligodendroglial cells as well.

**KEYWORDS**
brain-derived neurotrophic factor, chondroitin sulfate proteoglycans, glial cell line-derived neurotrophic factor, neurotrophic factor 3, oligodendrocyte progenitor cells, RRID:AB_1157905, RRID:AB_2535847, RRID:AB_357617, RRID:CVCL_0154, RRID:RGD_1566440

**Abbreviations:** BDNF, brain-derived neurotrophic factor; CNTF, ciliary neurotrophic factor; CSPG, chondroitin sulfate proteoglycan; GAG, glycosaminoglycan; GDNF, glial cell line-derived neurotrophic factor; NT, neurotrophic agent; NT-3, neurotrophic factor 3; OLs, oligodendrocytes; OPCs, oligodendrocyte progenitor cells; SCI, spinal cord injury.

Edited by Christina Ghiani. Reviewed by Alexander Gow and Friederike Pfeiffer.

- - - - - - - - - - - - - - - - - - - - - - - - - - - - - - - - - - - - - - - - - - - - - - - - - - - - - - - - - -

# 1 | INTRODUCTION

Damage to neurons and axonal pathways is largely credited for the loss of sensory and motor function that follows injury to the spinal cord (SCI). Furthermore, demyelination of neighboring intact axons will contribute to and potentially exacerbate the loss of function. Repairing damaged axonal pathways has proved elusive, as the central nervous system is notoriously known for its limited and/or incomplete regenerative response (reviewed by Cregg et al., 2014; Fawcett, 2020; Silver et al., 2015). This is largely attributed to the post-injury response of the lesioned spinal cord, which includes microglial activation, vascular macrophage infiltration, myelin breakdown, and formation of a glial scar, all of which contribute to the creation of a localized environment hostile to axonal regeneration and remyelination (Fawcett, 2020; Galtery & Fawcett, 2007; Porfyris et al., 2004; Siebert et al., 2014, 2015).

It follows then, that treating the area of a spinal lesion with agents that promote neuronal survival and axonal regeneration has been a major focus of spinal cord research aimed at restoring functional recovery. One promising avenue of research involves treating the spinal lesion site with the bacterial enzyme chondroitinase ABC (cABC), which neutralizes the inhibitory effects of chondroitin sulfate proteoglycans (CSPGs) present in the glial scar. Many studies demonstrate a significant improvement in axonal sprouting and regenerative growth with cABC treatment (reviewed by Bradbury & Burnside, 2019; Cregg et al., 2014; Siebert et al., 2015). Another treatment strategy utilizes neurotrophic agents to promote neuronal survival and stimulate the regeneration of axons. Currently some of the most common neurotrophic agents utilized in spinal regeneration studies include brain-derived neurotrophic factor (BDNF; Gransee et al., 2015; Hassannejad et al., 2019; Liu et al., 2017), glial cell line-derived neurotrophic factor (GDNF; Chen et al., 2018; Haldeman & Manna, 2019; Tajdaran et al., 2016), and neurotrophic factor-3 (NT-3; Han et al., 2019; Li et al., 2016; Xu et al., 2020). In these studies, the application of neurotrophic factors can enhance neuron survival and axonal sprouting, depending on the neuronal population examined and the SCI model.

The local environment of a spinal lesion is also known to influence the behavior of endogenous oligodendrocyte progenitor cells (OPCs) in the spinal cord. Recent literature demonstrates CSPGs are highly inhibitory to OPCs, impairing their migration to the lesion site, and terminal differentiation (Harlow & Macklin, 2014; Keough et al., 2016; Lau et al., 2012; Pendleton et al., 2013; Siebert & Osterhout, 2011; Siebert et al., 2011). Similar to the neuronal regeneration studies, treatment of an SCI lesion site with cABC allows the endogenous OPCs to migrate into the area of the lesion and differentiate (Keough et al., 2016; Siebert & Osterhout, 2011; Siebert et al., 2011). Neurotrophins have been shown to enhance remyelination in animal models of demyelination, and can enhance Schwann cell myelination after SCI (Chen et al., 2018; Mekhail et al., 2012; Zhang et al., 2009). In the present study, we investigated whether specific neurotrophic (NT) agents, such as BDNF, CNTF, GDNF, and NT-3, would have effects on the behavior of endogenous OPCs near a spinal lesion. Using an *in vitro* model of the glial scar, the findings demonstrate that certain neurotrophins can neutralize the inhibitory

**Significance**

Following a spinal cord injury, specific molecules expressed around the lesion are highly inhibitory to neuronal regeneration and remyelination by endogenous oligodendrocyte progenitor cells (OPCs). Potential treatments include neurotrophins that can promote neuronal survival and stimulate regeneration. In this study, neurotrophins that promote axonal growth were tested for their effects on OPCs maintained on the molecular components of a glial scar. The results show that certain neurotrophins will stimulate process outgrowth and differentiation of OPCs, even in the presence of these inhibitory molecules. The findings suggest that neurotrophins may work to promote both axonal regeneration and remyelination after injury.

influences of CSPGs (without the use of cABC) and stimulate OPC process outgrowth; an essential step in the myelination of axons.

# 2 | MATERIALS AND METHODS

All experiments requiring the use of animals were approved by the SUNY Upstate Medical University Committee for the Humane Use of Animals, following the provisions and guidelines set forth by the Department of Laboratory Animal Resources and the Association for Assessment and Accreditation of Laboratory Animal Care.

## 2.1 | Cell culture

### 2.1.1 | Primary cell culture

Primary mixed glial cultures were isolated from neonatal (P2) Sprague–Dawley rats (Taconic Farms; RRID:RGD_1566440) as previously described (Osterhout et al., 1997). As these primary mixed glial cultures were obtained from entire litters of neonatal rat pups, the glial cultures were created from a mix of male and female animals. In brief, brains were removed and the cortices disassociated into a mixed glial suspension. Glial cultures were grown in T-75 cm$^2$ tissue culture flasks (Corning) until the formation of a confluent astrocytic monolayer. OPCs were harvested using a mitotic shake off and purified by several differential plating techniques. OPCs were maintained in a defined, serum free media containing insulin, selenium, transferrin, and triiodothyronine, and supplemented with either B104 neuroblastoma conditioned media (B104-CM, 30%v/v; RRID:CVCL_0154) or a mix of PDGF-AA (10 ng/ml) and FGF2 (20 ng/ml) growth factors (PeproTech) to prevent OPC differentiation. The purity of each primary OPC preparation was assessed prior to utilization in experiments by staining with A2B5 and/or PDGFRα. The cultures were routinely at least 95% OPCs, with no contamination by microglia, and very few astrocytes.

## 2.1.2 | *In vitro* glial scar

This study utilized an *in vitro* model to mimic the glial scar previously described (Siebert & Osterhout, 2011). Prior to cell plating, glass coverslips were coated in a commercially available mix of CSPGs (CC117; MilliporeSigma) combined with laminin to aid cell adhesion. This particular mix of CSPGs has been shown to be highly representative of the CSPGs found within the glial scar (Monnier et al., 2003) Coverslips were set up in 24 well plates (Corning) and first coated overnight in poly-lysine (Sigma). Coverslips were rinsed with PBS and subsequently coated in either laminin (10 mg/ml; Invitrogen) or 50 μg/mL of CSPGs mixed in laminin.

## 2.2 | Experimental conditions

### 2.2.1 | Growth factor assays

Purified OPCs were plated on precoated coverslips at a density of 2,000 cells per coverslip, and maintained in a defined, serum free media containing insulin, selenium, apo-transferrin, and triiodothyronine (Osterhout et al., 1999) supplemented with B104-CM (30%v/v) or mix of PDGF-AA (10 ng/ml) and FGF2 (20 ng/ml) growth factors (PeproTech) to prevent the start of OPC differentiation. Twenty-four hours post-plating, the culture media was changed to a defined differentiation media. This media composition is identical to the serum free plating media described above, except it lacks B104CM, or the PDGF/FGF mix. Removal of these factors will initiate OPC differentiation in culture. When differentiation media is added, the cells are treated with either saline, BDNF, CNTF, GDNF, or NT-3 (PeproTech) to a final concentration of 10, 25, or 50 ng/ml. Cell cultures were re-fed every other day with fresh media and growth factors. An initial coverslip was fixed prior to initiation of treatment, and cultures were fixed at 24, 48, and 72 hr post-treatment.

## 2.3 | Immunocytochemistry

### 2.3.1 | Growth factor assay

Coverslips taken at 24 and 48 hr time points were live stained for 30 min at room temperature with A2B5 antibody (Table 1; Hybridoma bank; RRID:AB_1157905), while those taken at 72 hr

post-treatment were live stained with O4 antibody (Table 1; R&D Systems; Catalogue: MAB1326; RRID:AB_357617). Cells were then fixed in 4% paraformaldehyde in 1xPBS (pH 7.4) for 30 min, and stained using the Goat-anti-mouse-igM Alexa Fluor 555 (Table 1; Molecular Probes; Catalog # A-21426; RRID:AB_2535847) at a 1:500 dilution mixed in 1xPBS (pH 7.4) for 1 hr at room temperature. Coverslips were mounted on slides using Prolong Gold mounting media containing DAPI (Molecular Probes).

### 2.3.2 | Antibody validation

Negative controls were run for all antibody staining experiments. They included incubating the cells in secondary antibodies, without any prior exposure to the primary antibody, which resulted in no visible staining. Additional negative controls included incubation of the primary IgM antibody with a secondary anti-IgG antibody, again resulting in no visible staining. Further confirmation of antibody specificity included the use of DAPI in the mounting media, which clearly indicated the presence of other glial cells on the coverslips (astrocytes and microglia) which did not stain with either the primary or secondary antibodies. Furthermore, all positive, visible staining followed expected cellular morphological patterns.

## 2.4 | Data analysis

### 2.4.1 | Imaging

All images were taken on a Zeiss Axio Imager A1 (Zeiss, Germany) using a SPOT model 2.3.1 camera (Diagnostic Instruments). Overall fields of view were imaged using a 16× water immersion objective (Zeiss, Germany), while high magnification images used for cell counts, were visualized using a 40× objective (Zeiss, Germany), for a clear view of cell morphology. All images were edited using the SPOT imaging software to uniformly adjust brightness and contrast. No other manipulations to the micrographs were made.

### 2.4.2 | Growth factor assay analysis

All experiments were performed using two separate preparations of primary OPCs. In each experiment, there were three coverslips

**TABLE 1** Antibody information

| Name | Description of immunogen | Source, host species, Cat No., RRID | Concentration used |
|---|---|---|---|
| Glial precursor marker, A2B5 | Full-length sequence | Developmental studies hybridoma bank; mouse monoclonal, RRID:AB_1157905 | 5 μg/ml; IFC |
| Human/mouse/rat/chicken oligodendrocyte marker O4 antibody | Bovine brain corpus callosum white matter | R&D systems; mouse monoclonal, MAB1326; RRID:AB_357617 | 10 μg/ml; IFC |
| Goat anti-mouse IgM, Alexa flour 555 | Mouse mu immunoglobulin | Invitrogen; Goat, A-214226; RRID:AB_2535847 | 1:500; IFC |

for each time point and growth factor treatment. Additionally, every experiment included a treatment control in which the OPCs were plated on 50 µg/mL CSPGs and treated with differentiation media supplemented with saline. The total number of OPCs were counted for all conditions, and scored for the degree of process outgrowth and state of differentiation. The percent of cells exhibiting process outgrowth was determined by dividing the total number of immunolabeled OPCs extending at least two processes, longer than a cell body in length, by the total number of immunolabeled cells in the entire field of view. The number of cells extending processes was further classified into: (a) Simple process outgrowth, defined as an OPC extending less than four primary processes from the

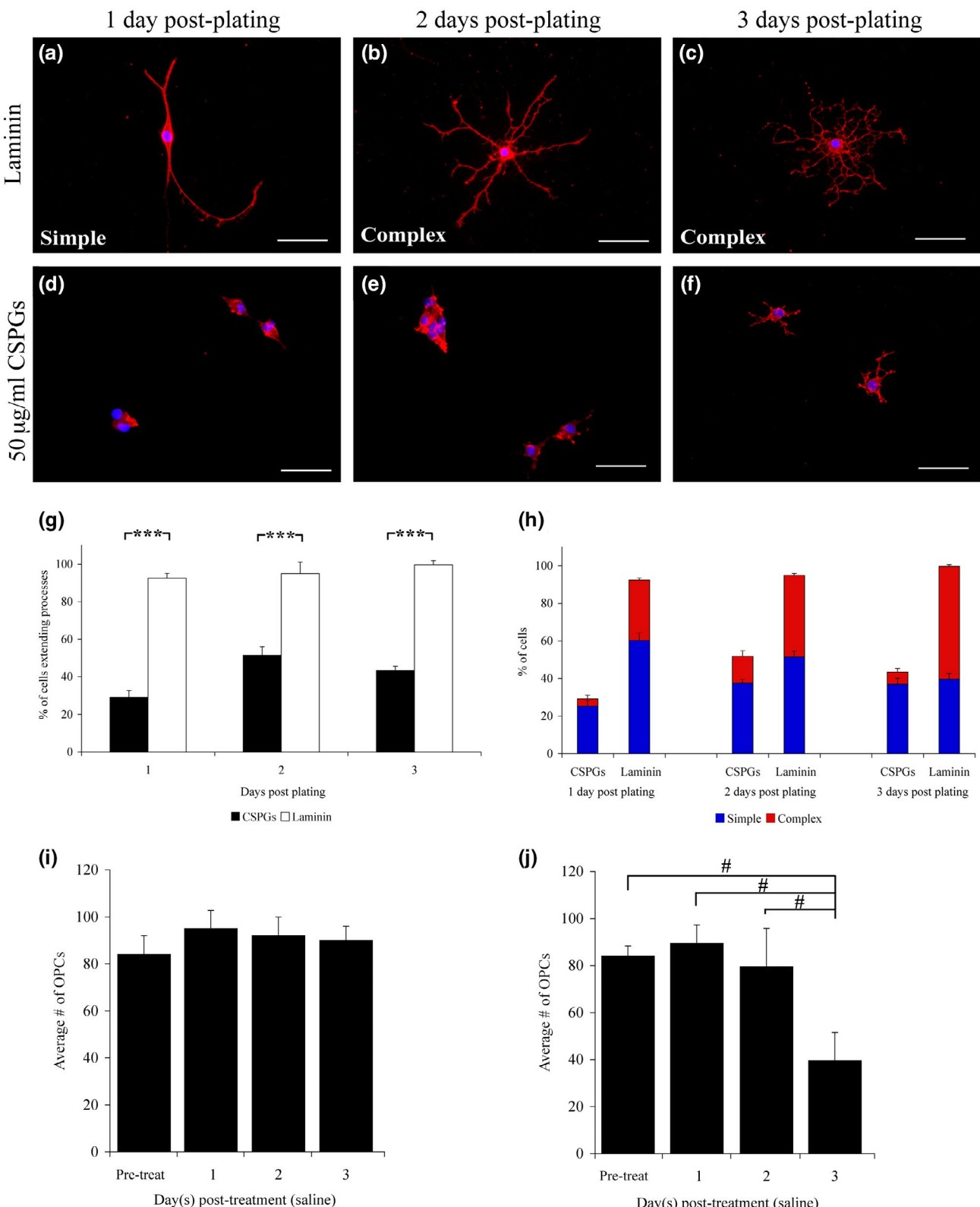

**FIGURE 1** Morphological and quantitative analysis of OPCs. One day after plating, OPCs on laminin exhibit simple bipolar process outgrowth (a). In contrast, OPCs on laminin with 50 µg/ml CSPGs extended short, highly stunted processes (d). At both 2 and 3 days post-plating, a majority of the OPCs on laminin demonstrated complex process outgrowth (b,c), while those plated on CSPGs remained unable to extend long and complex processes (e,f). Quantification of the percent of OPCs extending processes revealed a significant ($p \leq 0.001$; Student's *t* test) decrease in the number of cells with processes on CSPGs as compared with laminin at all three time points examined (g). OPCs plated on CSPGs exhibit primarily simple, short, and unbranched process outgrowth, while those plated on laminin undergo increasing levels of complex, branched process outgrowth (h). Quantification of the average number of OPCs plated on laminin, at each time point, revealed a consistent number of cells on the coverslips (i). There was a significant decrease ($p \leq 0.05$) in the average number of OPCs found at 3 days post-plating in the OPCs plated on 50 µg/ml CSPGs (j). All OPCs were visualized at 1 and 2 days post-plating with A2B5 immunoreactivity and at 3 days with O4 immunohistochemistry. Scale bar = 100 mm, and error bars = *SEM*, #$p \leq 0.05$ significance level, ***$p \leq 0.001$ significance level [Color figure can be viewed at wileyonlinelibrary.com]

cell body, each longer than one cell body in length, and have little to no secondary processes branching off the primary cytoplasmic branches (Figure 1a), or (b) Complex process formation, defined as four or more primary cytoplasmic processes branching off of the cell body, longer than a cell body in length, with numerous secondary processes branching off of the primary cytoplasmic cell processes (Figure 1b,c). Cells had to be a minimum distance of two cell bodies (diameter) apart from each other to be scored. A total of 20 visual fields were counted on each coverslip. Statistical analysis was performed using Microsoft Excel running a two-way ANOVA (Time and Neurotrophin Concentration) to test for significant differences in our treatment groups, followed by a Tukey's post hoc test, to determine the significance between individual treatments. Significance was identified to at least the $p = 0.05$ level.

### 2.4.3 | Cell death and proliferation

The average total number of OPCs counted for each time point of a treatment condition was determined to reveal any change in cell number due to cell death or proliferation. Statistical analysis was performed using Microsoft Excel running a two-way ANOVA to test for any significant differences in average total number of OPCs counted at each time point for a treatment, followed by a Tukey's post hoc test, to determine the significance between individual time points within a treatment. Significance was identified to at least the $p = 0.05$ level.

## 3 | RESULTS

### 3.1 | CSPGs significantly inhibit OPC process outgrowth

OPC process extension is an essential initial step in the differentiation of OPCs into mature myelinating oligodendrocytes. Progenitor cells have a simple bipolar morphology; as they begin to differentiate, they extend multiple processes that become more branched and complex over time. This is evident in the OPC cultures maintained on laminin, which quickly form processes as they differentiate over 3 days *in vitro* (Figure 1). Initially after plating, almost all OPCs (92%) extended processes; of these, a majority (60%) showed a simple

bipolar morphology (Figure 1a,h). Twenty-four hours post-plating, a defined media was added to initiate OPC differentiation. At 2 days post-plating, (24 hr after the media change) approximately 50% (Figure 1h) of the cells displayed a more complex morphology, typically observed at the early stages of OPC differentiation (Figure 1b). By 3 days post-plating, all the observed cells extended processes, and the majority (65%) showed extensive process outgrowth, with many intricate and highly branched processes (Figure 1c,h).

The addition of neurotrophins to control cultures (OPCs on laminin) did not influence the time course of process outgrowth. On laminin, the differentiation of OPCs proceeds quickly (Figure 1), with noticeable morphological changes evident daily. While neurotrophins such as CNTF and NT-3 may stimulate the differentiation of OPCs plated on substrates such as polylysine (Barres et al., 1996; Du et al., 2003; Heinrich et al., 1999; Yan & Wood, 2000), there were no identifiable or significant effects observed with the addition of any of the growth factors on the time course of OPCs differentiating on laminin (data not shown).

In contrast, the addition of CSPGs has a dramatic effect on OPC behavior. OPCs maintained on the 50µg/mL CSPG substrate showed a significant ($p \leq 0.001$) decrease in process outgrowth at all time points (Figure 1d,e,f; Siebert & Osterhout, 2011). Only 25% of OPCs (Figure 1g) extended cellular processes longer than one cell body in diameter 1 day post-plating. When process outgrowth was examined at both 2 days (51%) and 3 days (43%), a significant reduction (Figure 1g; $p \leq 0.001$) in OPC process outgrowth was observed. Additionally, there was also a marked difference in the morphology of the OPCs. Cells on the CSPG substrate tended to grow in aggregates (Figure 1d,e), while those plated on laminin did not (Figure 1a,b). At the three-day time point, when most OPCs are differentiating on laminin, the morphology of any processes extending from OPCs on CSPGs was very short and stunted, with minimal branching (Figure 1d–f). Those OPCs plated on laminin produce an extensive and complex network of branched processes over 3 days (Figure 1a–c,h).

The number of OPCs plated on laminin did not change over the course of the experiment, with no significant difference in the average number of cells (Figure 1i). However, there was a significant reduction (Figure 1j; $p \leq 0.05$) in the average number of OPCs maintained on CSPGs between 2 and 3 days post-plating. We did not observe apoptotic cells with condensed nuclei at these time points, but an increase in floating cells was noted. Most of the

cell loss is due to poor attachment, with OPCs detaching from the substrate with media changes and handling of the culture dishes. If the floating OPCs were replated on a laminin substrate, they attached and extended normal processes, indicating their viability (data not shown).

## 3.2 | BDNF treatment promotes OPC process outgrowth in the presence of CSPGs

BDNF is a neurotrophic molecule that has been utilized to increase neuronal survival and axonal regeneration following SCI

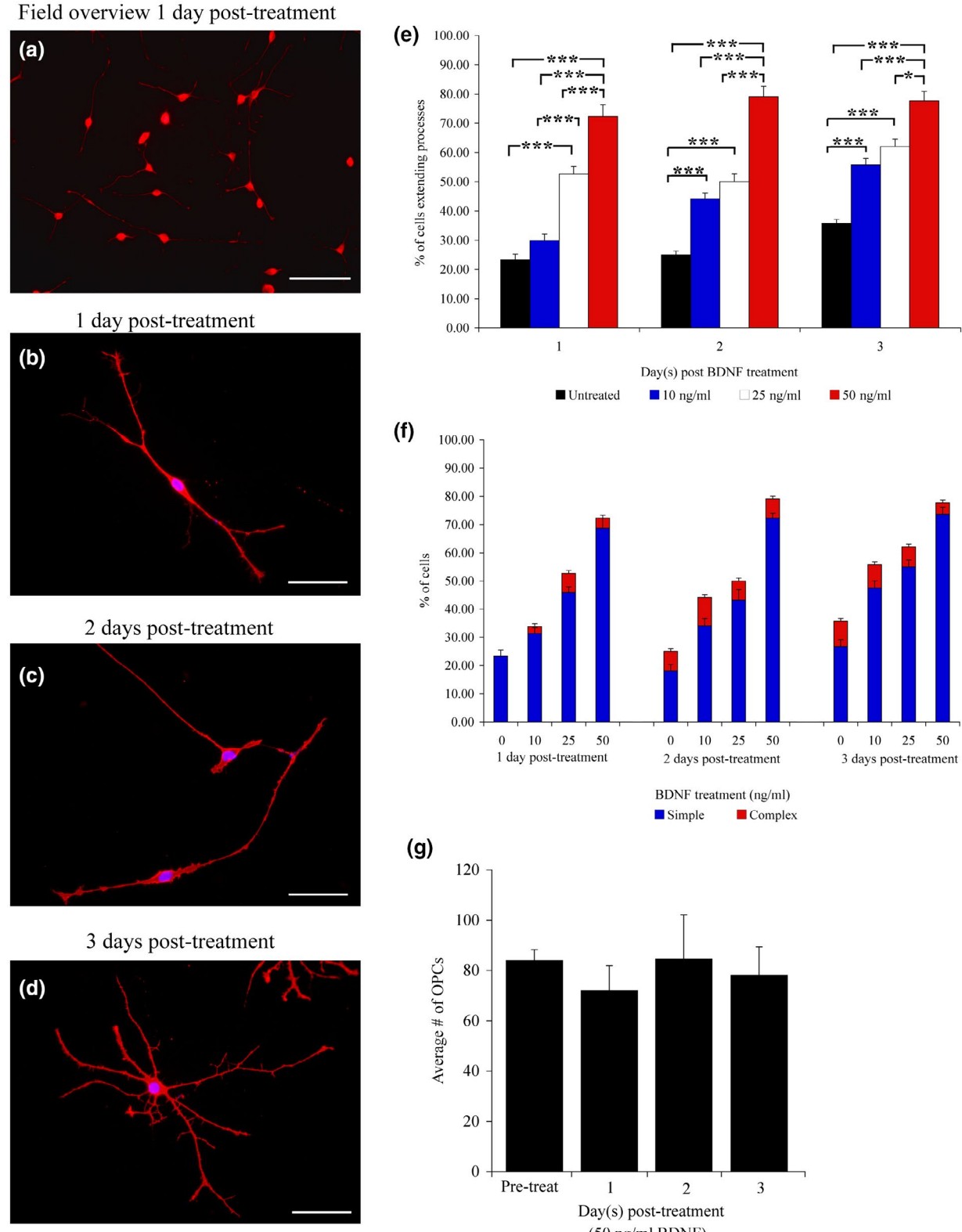

**FIGURE 2** Morphological and quantitative analysis of OPCs treated with BDNF. A low power overview reveals that treatment with 50 ng/ml BDNF stimulates process outgrowth from OPCs plated on the laminin–CSPG mix. (a) At higher magnification, the long processes are easily observed at 1, 2, and 3 days post-plating (b–d). At 1 and 2 days after BDNF addition, the cells have long bipolar processes (b,c); at 3 days, it appears they are beginning to differentiate (d). When the effects of BDNF were quantified, significant differences were observed in all three time points examined (1 day post-treatment = $F(3, 156)$ = 5.59; $p ≤ 0.001$; 2 days post-treatment = $F(3, 156)$ = 5.15; $p ≤ 0.001$; and 3 days post-treatment = $F(3, 156)$ = 4.58; $p ≤ 0.01$) (e). Quantification of process complexity demonstrates that BDNF stimulates the extension of primarily long simple unbranched processes at all three time points (f). There is no difference in the average number of OPCs counted per time point for BDNF or laminin controls (g). OPCs were visualized at 1 and 2 days post-plating with A2B5 immunoreactivity, and 3 days with O4 immunohistochemistry. Scale bars = 300 mm (a), 100 mm (b–d), and error bars = *SEM*, *$p ≤ 0.01$ significance level, ***$p ≤ 0.001$ significance level [Color figure can be viewed at wileyonlinelibrary.com]

**TABLE 2** BDNF treatment OPC data quantification

| % of OPCs extending processes; BDNF treatment | | | | | | |
|---|---|---|---|---|---|---|
| | **Day 1** | | **Day 2** | | **Day 3** | |
| **Treatment** | **#/N** | **% (SD)** | **#/N** | **% (SD)** | **#/N** | **% (SD)** |
| Untreated | 46/197 | 23.35 (1.92) | 40/160 | 25.01 (1.34) | 40/112 | 35.71(1.40) |
| 10 ng/ml | 70/207 | 33.81 (2.16) | 57/129 | 44.18 (1.91) | 68/122 | 55.73 (2.24) |
| 25 ng/ml | 108/205 | 52.68 (2.57) | 103/206 | 50.00 (2.74) | 98/158 | 62.02 (2.62) |
| 50 ng/ml | 141/195 | 72.30 (4.04) | 117/148 | 79.05 (3.62) | 115/148 | 77.70 (3.17) |

(Gransee et al., 2015; Hassannejad et al., 2019; Liu et al., 2017). BDNF has been shown to markedly increase DNA synthesis and overall cell numbers when applied to OLs, along with an increase in OPC maturation (Du et al., 2006; Van't Veer et al., 2009). Furthermore, it has been shown that following a SCI, local glial cells (astrocytes, microglia) and macrophages express BDNF, so endogenous OPCs may be exposed to it in the local microenvironment (Dougherty et al., 2000). Therefore, we examined if the beneficial growth promoting effects of BDNF could reverse the observed inhibition of OPCs maintained in the presence of CSPGs in our culture model.

At higher concentrations of BDNF (25 or 50ng/ml), there was a significant increase in the number of OPCs exhibiting simple process outgrowth observed at all time points (Figure 2a–e; Table 2; $p ≤ 0.001$). At the lowest concentration (10ng/ml) of BDNF, there was no significant difference in process outgrowth between untreated controls and BDNF treated cells in the first 24 hr. However, there was a gradual increase in the number of OPCs extending processes at both 2 and 3 days post-plating, (Figure 2e; Table 2; $p ≤ 0.001$). There is a clear effect of BDNF concentration, with the percent of OPCs exhibiting process outgrowth being highest with 50 ng/ml BDNF, as compared to lower concentrations at all three time points (Figure 2e). Process outgrowth was slower to proceed at the lowest concentration (day 1; 10 ng/ml) but caught up by the second day of treatment (Figure 2e). Closer examination of the cellular morphology of the BDNF treated cells demonstrated that for all 3 days post-treatment examined, and for all three concentration treatment groups (10, 25, 50 ng/ml) the majority of OPCs were found to exhibit a simple and bipolar cellular morphology (Figure 2f). Furthermore, BDNF did not appear to stimulate either OPC proliferation or cell death, as comparison of the overall number of OPCs observed at all time points did not vary significantly from each other (Figure 2g).

## 3.3 | GDNF treatment promotes OPC process outgrowth in the presence of CSPGs

GDNF is a neurotrophic factor with a demonstrated ability to enhance the axonal sprouting and regeneration of neurons following axotomy (Chen et al., 2018; Haldeman & Manna, 2019; Tajdaran et al., 2016). Interestingly, GDNF has also been found to be expressed in a spinal contusion lesion by immune cells at early times post-injury (Hashimoto et al., 2004; Satake et al., 2000). GDNF was therefore included as one of the neurotrophins tested for its effects on OPCs maintained in the presence of CSPGs *in vitro*.

Treatment of OPCs with GDNF restored process outgrowth in as early as 24 hr post-treatment, as shown in Figure 3a–e and Table 3. At the highest concentration (50 ng/ml), the majority of OPCs showed simple bipolar processes 1-day post-GDNF treatment (Figure 3f; Table 3). At later time points, process outgrowth continued and the OPCs started to show signs of differentiation with the appearance of complex, branched processes 3 days post-treatment (Figure 3d,f). The effects of GDNF were evident at lower concentrations, with a significant increase in the number of OPCs exhibiting process outgrowth observed in the OPCs treated with 10, 25, or 50 ng/ml GDNF, compared to the untreated controls, at all three time points (Figure 3e; $p ≤ 0.001$).

There was a clear differential effect of GDNF concentration observed on the first day of treatment. At 1-day post-GDNF treatment, there was a significant ($p ≤ 0.01$) increase in the number of OPCs with measurable process outgrowth, correlating with the increasing concentrations of GDNF (10, 25 and 50 ng/ml). However, at both 2 and 3 days post-plating, there was no difference in the percent of OPCs extending processes between OPCs treated with 10 and 25 ng/ml GDNF (Figure 3e). At all three time points, a majority of

the cells were simple and bipolar (Figure 3f). However, those OPCs treated with the highest dose (50 ng/ml) showed a more robust response. Not only did more cells extend processes, but also at later time points, it appeared that the cells started to differentiate, showing more complex process outgrowth (Figure 3f). Furthermore, GDNF did not appear to stimulate either OPC proliferation or cell death, as comparison of the overall number of OPCs observed at all times did not vary significantly (Figure 3g).

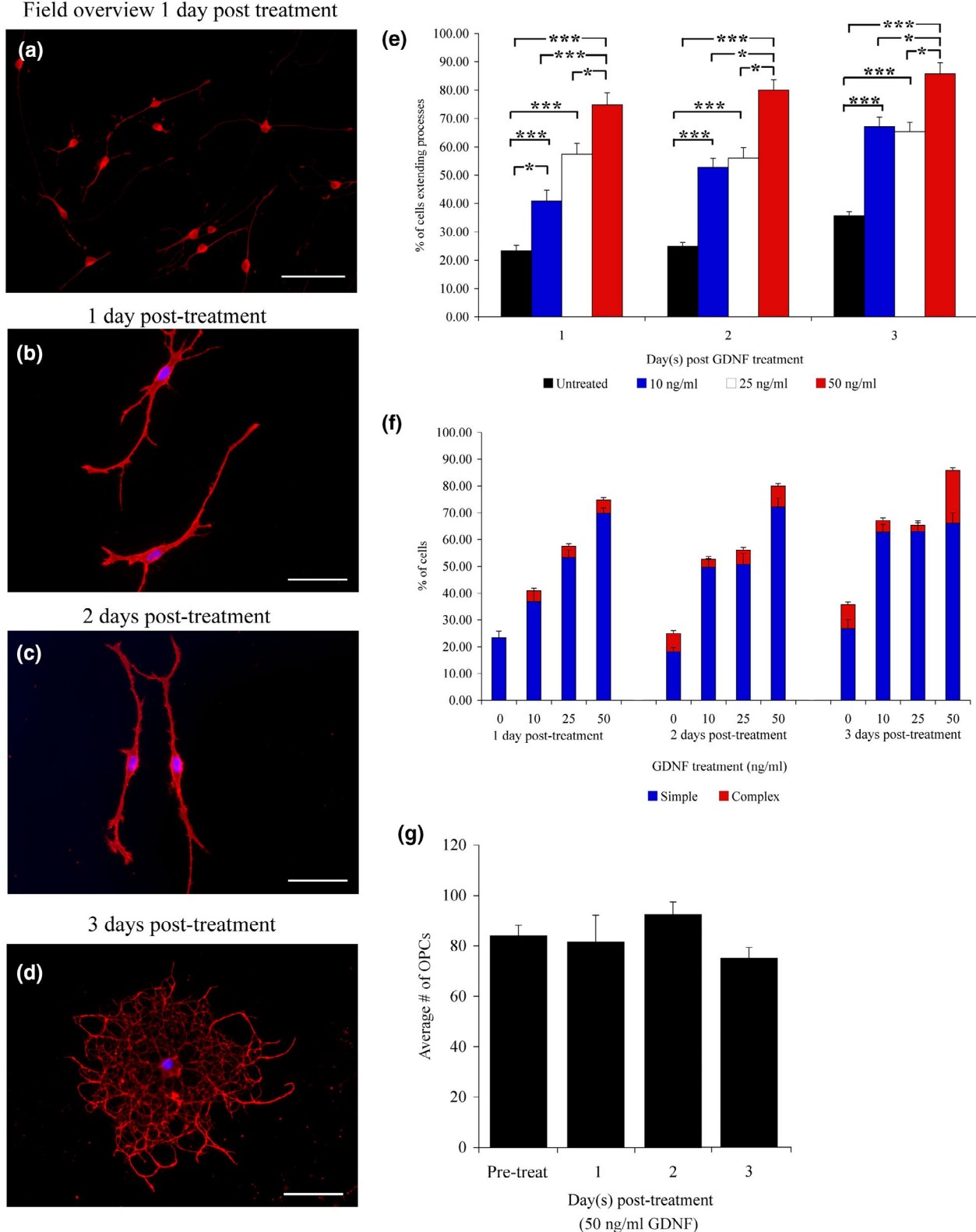

**FIGURE 3** Morphological and quantitative analysis of OPCs treated with GDNF. A low power overview reveals that treatment with 50 ng/ml GDNF neutralizes the inhibition by CSPGs. Many cells are extending processes after only 1 day of GDNF treatment while being maintained on 50 µg/ml CSPGs (a). Long bipolar processes are evident after GDNF treatment at 1 and 2 days post-treatment (b,c). At 3 days post-GDNF treatment, OPCs initiated more complex, highly branched processes, and appear to be differentiating (d). When effect the effect of GDNF treatment on OPC process outgrowth was quantified, significant differences were observed in all three time points examined (1 day post-treatment = $F_{(3, 156)}$ 4.27; $p \leq 0.01$; 2 days post-treatment = $F_{(3, 156)}$; $p \leq 0.01$; and 3 days post-treatment = $F_{(3, 156)}$; $p \leq 0.01$) (e). Regardless of GDNF concentration, OPCs mainly extend simple processes at all three time points examined (f). Quantification of the average number of OPCs counted per time point shows no significant changes in OPC numbers (g). OPCs were visualized at 1 and 2 days post-plating with A2B5 immunoreactivity, and 3 days with O4 immunohistochemistry. Scale bars = 300 mm (a), 100 mm (b–d), and error bars = *SEM*, *$p \leq 0.01$ significance level, ***$p \leq 0.001$ significance level [Color figure can be viewed at wileyonlinelibrary.com]

**TABLE 3** GDNF treatment OPC data quantification

| % of OPCs extending processes; GDNF treatment | | | | | | |
|---|---|---|---|---|---|---|
| | Day 1 | | Day 2 | | Day 3 | |
| Treatment | #/N | % (SD) | #/N | % (SD) | #/N | % (SD) |
| Untreated | 46/197 | 23.35 (1.92) | 40/160 | 25.01 (1.34) | 40/112 | 35.71(1.40) |
| 10 ng/ml | 101/247 | 40.89(3.86) | 107/203 | 52.70 (3.31) | 110/164 | 67.07 (3.41) |
| 25 ng/ml | 139/242 | 57.43 (3.86) | 106/189 | 56.08 (3.68) | 115/176 | 65.34 (3.36) |
| 50 ng/ml | 151/202 | 74,75 (4.34) | 132/165 | 80.00 (3.62) | 127/148 | 85.81 (3.90) |

## 3.4 | NT-3 treatment stimulates OPC process outgrowth and differentiation in the presence of CSPGs

NT-3 has also been shown to enhance neuronal survival and stimulate regenerative sprouting following SCI (Han et al., 2019; Li et al., 2016; Tobias et al., 2003; Xu et al., 2020). Moreover, it can enhance the survival of OPCs, both *in vitro* and *in vivo* (Kumar et al., 1998). In this experiment, we examined the effect of NT-3 on OPCs plated on CSPGs *in vitro*.

Similar to the other growth factors, NT-3 also neutralized the inhibitory effects of CSPGs on process extension from OPCs. Unlike the effects of BDNF and GDNF, a larger percent of the OPCs exhibited the morphology of differentiating cells 24 hr after the addition of the highest concentration of NT-3 (50 ng/ml; Figure 4a,b; Table 4). At longer times post-treatment, the cells displayed complex process outgrowth, with many highly branched processes (Figure 4c,d). The initial start of membrane sheet formation can be observed at 3 days after NT-3 addition (Figure 4d). Even at lower concentrations of NT-3 (10, 25, or 50 ng/ml), there was a highly significant increase in the percentage of OPCs extending processes at all time points (Figure 4e; Table 4; $p \leq 0.001$). The effect of NT-3 concentration on OPCs was most apparent at 1 and 2 days post-treatment, where significant differences were found between each concentration (Figure 4e). However, at 3 days after treatment, the extent of process outgrowth was equivalent at either 25 or 50 ng/ml of NT-3. Significant differences in OPC morphology was observed, with the majority of the cells showing extensive highly branched processes (Figure 4c,d,f). NT-3 treatment did not appear to stimulate either the proliferation or death of OPCs, as the numbers of cells did not significantly vary at any time (Figure 4g).

## 3.5 | CNTF has no observable effect on OPC process outgrowth in the presence of CSPGs

Several studies have demonstrated that CNTF enhances the rate of OL generation, long-term survival, and promotes the maturation of OLs (Barres et al., 1993, 1996; Dell'Albani et al., 1998; Mayer et al., 1994). Our control studies did not show any effect of CNTF addition, albeit in this series of experiments, OPCs are grown on laminin, and laminin itself is growth promoting (Buttery & ffrench-Constant 1999; data not shown). CNTF has also been shown to enhance the survival, axonal sprouting, and regenerative growth of certain populations of neurons (Siegel et al., 2000; Ye et al., 2004; Ye & Houle, 1997). Given its documented effects on the oligodendroglial cells and its use as a potential treatment for SCI, CNTF was tested on OPCs maintained on CSPGs *in vitro*.

Surprisingly, OPCs in both controls and CNTF treated cultures were typically observed clumped in aggregates when treated with CNTF, at either 25 or 50 ng/ml (Figure 5a; data not shown), with a small number of processes extending from the cell clump. However, because the cells were growing in an aggregate, they had to be excluded from quantification, as cell density can influence cell behavior. When single OPCs were observed, they exhibited stunted process outgrowth. Even after CNTF treatment for 3 days, at both concentrations tested, the morphology of OPCs remained similar to control cultures. Quantitative analysis of single cells that met the scoring criteria showed there was no difference between the OPCs maintained on CSPGs in the presence or absence of CNTF (Figure 5b). This did not change over time, as at both 2 and 3 days post-CNTF treatment, the OPCs did not appear to respond to CNTF (Figure 5b and data not shown). Therefore, it appears that treatment with CNTF did not appear to have any effect on OPCs maintained in the presence of CSPGs.

## 3.6 | Effect of other growth factors on OPC process outgrowth in the presence of CSPGs

Finally, it is worth mentioning, that in the course of this study, all OPC cultures were maintained in serum free, defined culture media that was supplemented with either B104 CM (30% v/v), or 20 ng/ml

FGF and 10 ng/ml PDGF (See Material and Methods). During incubation in this defined media, there was no noticeable effect on OPC process outgrowth in the presence of CSPGs (Figure 1d,e,f). There was absolutely no difference in those cultures maintained in B104 CM or PDGF/FGF; neither condition would reverse the inhibition of growth observed in OPC cultures plated on CSPGs.

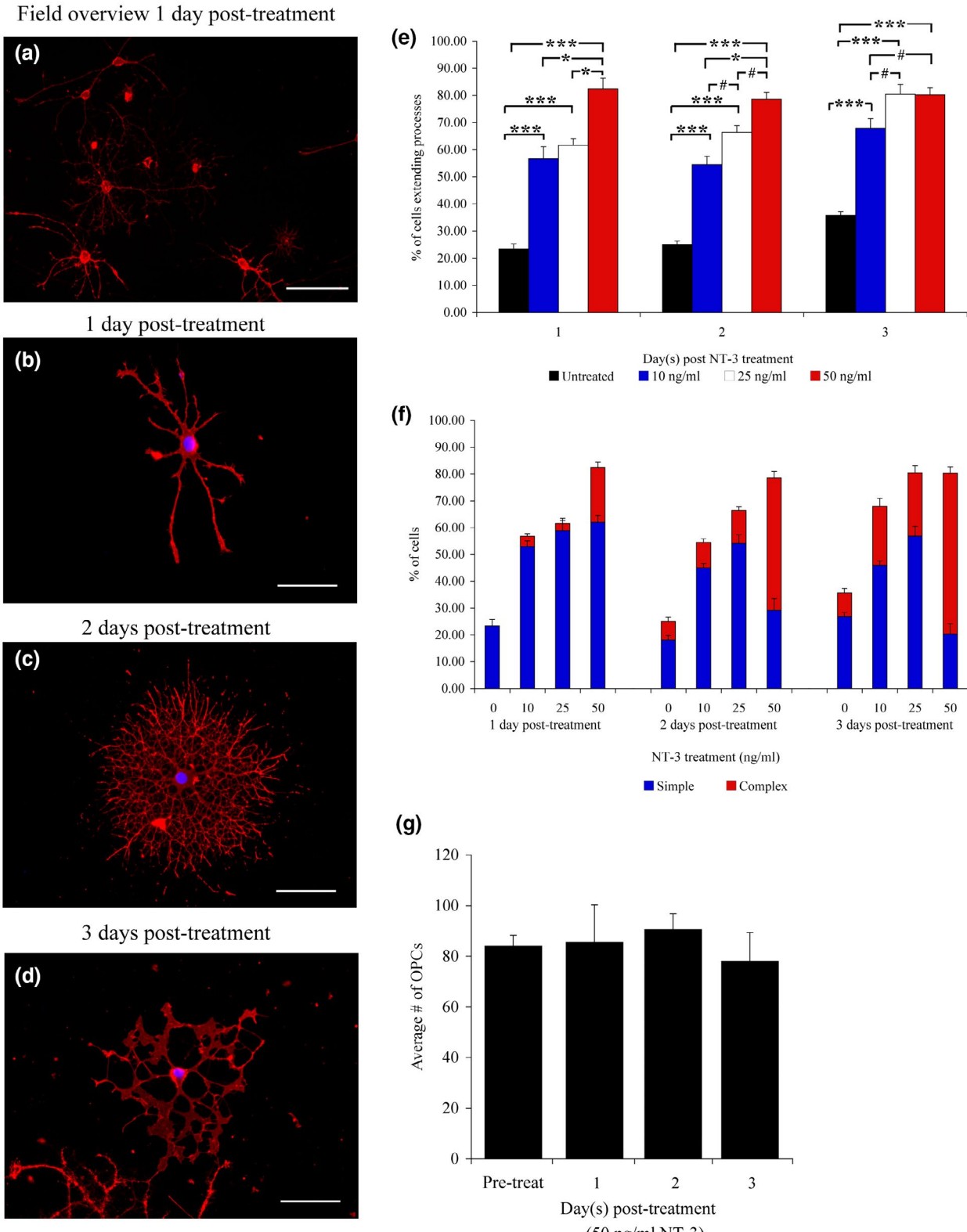

Field overview 1 day post-treatment

(a)

1 day post-treatment

(b)

2 days post-treatment

(c)

3 days post-treatment

(d)

(e)

Day(s) post NT-3 treatment

■ Untreated  ■ 10 ng/ml  □ 25 ng/ml  ■ 50 ng/ml

(f)

NT-3 treatment (ng/ml)

■ Simple  ■ Complex

(g)

Day(s) post-treatment
(50 ng/ml NT-3)

**FIGURE 4** Morphological and quantitative analysis of OPCs treated with NT-3. A low power overview reveals that treatment with 50 ng/ml NT-3 stimulates significant process outgrowth (a). OPCs extend long unbranched processes at 1 day post-treatment (b), while more complex process outgrowth is observed at 2 days post-treatment. NT-3 appears to drive their differentiation, even in the presence of CSPGs (c). At 3 days post-NT-3 treatment, it appears that the OPCs have begun to mature into a mature oligodendrocyte phenotype, with some cells starting to extend membrane sheets (d). When the effects of NT-3 treatment on process extension were quantified, significant differences were observed at all three time points examined (1 day post-treatment = $F(3, 156) = 8.23$; $p \leq 0.001$; 2 days post-treatment = $F(3, 156) = 7.79$; $p \leq 0.001$; and 3 days post-treatment = $F(3, 156) = 8.46$; $p \leq 0.001$) (e). Quantification of the complexity of process outgrowth demonstrates that as the concentration of NT-3 increases, the percentage of OPCs extending complex processes increases at every time point (f). No significant difference in the average number of OPCs counted per time point was observed (g). Cells were visualized at 1 and 2 days post-plating with A2B5 immunoreactivity, and 3 days with O4 immunohistochemistry. Scale bars = 300 mm (a), 100 mm (b–d), and error bars = SEM, #$p$ = 0.05 significance level, *$p \leq 0.01$ significance level, ***$p \leq 0.001$ significance level [Color figure can be viewed at wileyonlinelibrary.com]

**TABLE 4** NT-3 treatment OPC data quantification

| % of OPCs extending processes; NT-3 treatment | | | | | | |
|---|---|---|---|---|---|---|
| | Day 1 | | Day 2 | | Day 3 | |
| Treatment | #/N | % (SD) | #/N | % (SD) | #/N | % (SD) |
| Untreated | 46/197 | 23.35 (1.92) | 40/160 | 25.01 (1.34) | 40/112 | 35.71(1.40) |
| 10 ng/ml | 135/238 | 56.72 (4.3) | 110/202 | 54.45 (3.07) | 148/218 | 67.88 (3.60) |
| 25 ng/ml | 69/112 | 61.60 (2.41) | 146/220 | 66.36 (2.56) | 160/199 | 80.40 (3.65) |
| 50 ng/ml | 178/216 | 82.40 (3.99) | 132/168 | 78.57 (2.56) | 130/162 | 80.24 (2.60) |

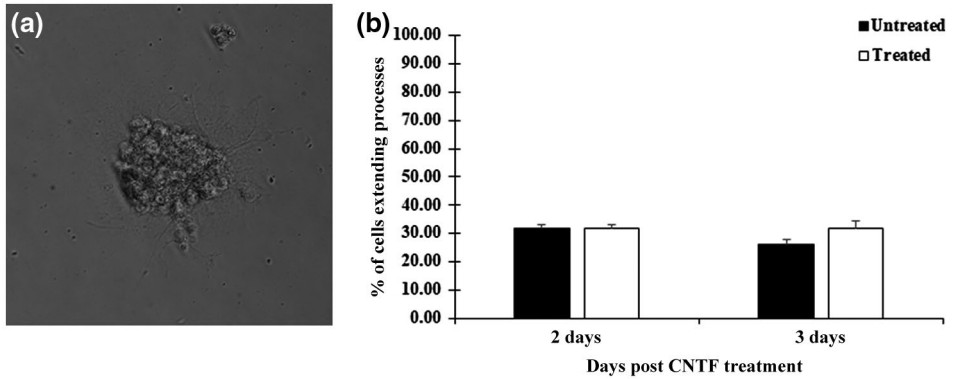

**FIGURE 5** Morphological and quantitative analysis of OPCs treated with CNTF. Phase contrast micrograph of an OPC cell aggregate growing on coverslips coated with laminin and 50 μg/ml CSPGs (a). OPCs growing in aggregation were excluded from quantitative analysis while OPC growing singularly could be quantified and demonstrated a lack of bipolar process outgrowth. There was no observable effect of CNTF treatment, as the percentage of OPC extending processes in the OPCs treated with CNTF (25 ng/ml) were very similar to those in the control conditions (b). Error bars = SEM

## 4 | DISCUSSION

The expression of CSPGs in and around a spinal cord lesion has been demonstrated to be highly inhibitory not only to the process of axonal regeneration but also to the migration of endogenous OPCs and their differentiation (Fawcett et al., 2012; Keough et al., 2016; Lau et al., 2012; Pendleton et al., 2013; Siebert & Osterhout, 2011; Siebert et al., 2011; Silver et al., 2015). Manipulation of the post-injury environment in models of SCI, by degrading the CSPGs with cABC, alone or in combination with neurotrophins can produce overall improvement of motor function (reviewed by DePaul et al., 2017; Griffin & Bradke, 2020). This improvement is largely attributed to promoting neuronal survival, and stimulating axonal sprouting and

regeneration (Blesch & Tuszynski, 2003, Iannotti et al., 2003; Lu et al., 2001; Tobais et al., 2003). Interestingly, the observed recovery of function does not correlate well with anatomical evidence of axonal regeneration and restoration of neuronal pathways (Bradbury & McMahon, 2006). However, there appears to be little consideration of potential influence of these factors on endogenous OPCs and the process of remyelination.

The present study revealed that neurotrophic agents used to enhance both axonal sprouting and neuronal survival following SCI would also promote process extension from OPCs. In most cases, the OPCs were able to extend processes on a CSPG substrate following treatment with BDNF, GDNF, and NT-3. Furthermore, the findings reveal differential effects of the NTs on the behavior of

OPCs in the presence of CSPGs. There is a high similarity in the effects of BDNF and GDNF, as both factors stimulate the formation of simple bipolar processes from OPCs to the same degree. OPCs cultured with GDNF gradually show signs of differentiating at 3 days after initial exposure; however, this was never observed with addition of BDNF. Treatment with BDNF restores the bipolar phenotype of OPCs plated on CSPGs, however, did not promote their morphological differentiation. In contrast, NT-3 treatment not only stimulated the OPCs to extend processes, but it appeared to drive their differentiation in the presence of CSPGs. OPCs treated with the NT-3 displayed a morphology of complex, branched processes with membrane sheet formation evident 3 days after NT-3 addition. The observed effects of NT-3 on OPCs in these experiments agree with previous work demonstrating that NT-3 treatment drives early differentiation of OPCs *in vitro* (Heinrich et al., 1999). The results suggest that the effect of NT-3 on OPCs is independent of the substrate.

In this study, we utilized an *in vitro* model mimicking the glial scar to test the effects of neurotrophic factors on the behavior of OPCs. The advantage of this model is that the OPCs are directly exposed to CSPGs without the influence of other inhibitory molecules, such as myelin breakdown products, found at the lesion site *in vivo*. This model allowed for the direct assessment of the effect of CSPGs on OPCs, and any subsequent effects of therapeutic agents on OPC behavior. The combination of laminin and CSPGs is sufficient to maintain OPC attachment and viability, even if the cells cannot extend normal processes. Therefore, this *in vitro* model of the glial scar is a useful system for rapid screening of any new biomolecules that may show promise in the treatment of SCI.

An unexpected finding was the inability of CNTF to stimulate process outgrowth from OPCs maintained on CSPGs. CNTF has been previously demonstrated to promote the growth and proliferation of OPCs *in vitro* (Barres et al., 1993, 1996; Dell'Albani et al., 1998; Mayer et al., 1994). A closer examination reveals that in these experiments, OPCs were plated on polylysine, which promotes cell attachment and serves as a permissive substrate, but does not actively stimulate process outgrowth. The current study utilized an *in vitro* model of the glial scar, with CSPGs that are commonly expressed after a SCI (Monnier et al., 2003). We have previously demonstrated that CSPGs will significantly inhibit process outgrowth, migration, and differentiation of OPCs, both *in vitro* and *in vivo* (Siebert & Osterhout, 2011; Siebert et al., 2011). Since CNTF did not have any stimulatory effects on OPCs in contact with CSPGs, these results suggest CSPGs have the ability to directly alter the OPCs ability to respond to CNTF.

This actually aligns with previous *in vivo* studies that demonstrate CNTF has little to no observable effect on OPCs in a spinal cord lesion. In a study by McTigue and colleagues (1998), fibroblasts were genetically engineered to secrete CNTF, and then implanted into the spinal cord following a SCI, in an attempt to drive remyelination of axons following injury. CNTF had no effect on the OPCs *in vivo*, and there was a lack of enhanced remyelination (McTigue

et al., 1998). This finding was confirmed in a study by Talbott and colleagues (2007). In this study, CNTF application to OPCs *in vitro* promoted OPC survival and maturation, however, when CNTF was applied to OPCs in the presence of reactive astrogliosis, resulting from an ethidium bromide lesion, CNTF had no observable effect on remyelination of lesioned axons (Talbott et al., 2007). Therefore, our observations suggest the presence of CSPGs may be one plausible explanation as to why CNTF is ineffective when utilized to treat OPCs in lesion *in vivo* (McTigue et al., 1998; Talbott et al., 2007). It is important to note that CNTF is not the only growth factor that has no effects on OPCs maintained on CSPGs. PDGF and FGF, both well-known mitogens for OPCs, did not induce proliferation when added to OPCs in the presence of CSPGs. The OPCs remained attached, but there was no increase in cell numbers and little process outgrowth in the presence of these growth factors (data not shown). Further studies are needed to elucidate how the CSPGs are potentially modulating the response of OPCs to growth factors, such as CNTF.

Collectively, these data indicate that certain neurotrophic factors, often utilized to promote axonal regeneration after SCI, also have a profound effect on the behavior of endogenous OPCs. More specifically, BDNF, GDNF, and NT-3, promote process outgrowth from OPCs and initiate the process of morphological differentiation in a CSPG rich environment. This discovery offers a possible explanation as to why significant recovery of motor function can be observed following spinal injury with little anatomic evidence of significant axonal regrowth (Bradbury & McMahon, 2006). Certain neurotrophins can neutralize the inhibitory effects of CSPGs in and around a spinal lesion, allowing for differentiation of endogenous OPCs. Ultimately, treatment of SCI with these factors may actually have a dual benefit, not only stimulating axonal sprouting, but also initiating remyelination of spared or regenerating axons to restore motor function.

## DECLARATION OF TRANSPARENCY

The authors, reviewers and editors affirm that in accordance to the policies set by the *Journal of Neuroscience Research*, this manuscript presents an accurate and transparent account of the study being reported and that all critical details describing the methods and results are present.

### ACKNOWLEDGMENTS
This work was supported by funding from the New York State Department of Health, Spinal Cord Injury Research Board grant # C020931 and Mentored Scientist grant # C022046, in addition to a grant from the Craig Neilsen Foundation awarded to Dr. Donna J. Osterhout.

### CONFLICT OF INTEREST
There is no conflict of interest for either JRS or DJO.

## AUTHOR CONTRIBUTIONS

Both J.R.S. and D.J.O. have full access to all the data in the study and are responsible for the integrity of the data and the accuracy of the data analysis. The first author named is the lead and corresponding author. We describe contributions to the paper using the CRediT taxonomy. *Conceptualization*, J.R.S. and D.J.O.; *Methodology*, J.R.S., D.J.O.; *Investigation*, J.R.S.; *Formal Analysis*, J.R.S.; *Writing – Original Draft*, J.R.S.; *Writing – Review & Editing*, J.R.S. and D.J.O.; *Project Administration*, D.J.O.; *Funding Acquisition*, D.J.O.

## PEER REVIEW

The peer review history for this article is available at https://publons.com/publon/10.1002/jnr.24780.

## DATA AVAILABILITY STATEMENT

Data sets regarding OPC counts are available from the authors upon reasonable requests.

## ORCID

*Justin R. Siebert* https://orcid.org/0000-0001-8653-2849

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

## SUPPORTING INFORMATION

Additional supporting information may be found online in the Supporting Information section.

Transparent Peer Review Report

Transparent Science Questionnaire for Authors

