## [Transparent Peer Review Report · Journal of Neuroscience Research]

Select Neurotrophins Promote Oligodendrocyte Progenitor Cell Process Outgrowth in the Presence of Chondroitin Sulfate Proteoglycans

Justin R. Siebert and Donna J. Osterhout

Review timeline:

Submission date: 27/04/20
Editorial Decision: Major Modification
(26-May-2020)
Revision Received: 30/09/20
Editorial Decision: Minor Modification
(18-Oct-2020)
Revision Received: 18/11/20
Editorial Decision: Accept with Minor
Edits (28-Nov-2020)
Revision Received: 01/12/20
Accepted: 07/12/20

Editor: Dr. Christina Ghiani
Reviewer 1: Alexander Gow
Reviewer 2: Friederike Pfeiffer

1st Editorial Decision

Dear Dr Siebert:

Thank you for submitting your manuscript to the Journal of Neuroscience Research. We've now received the reviewers' feedback and have appended those reviews below. As you will see, the reviewers find the question addressed to be of potential interest, however, some concerns were raised, hence, the manuscript is not suitable for publication in the current form.

If you feel that you can adequately address the concerns of the reviewers, you may revise and resubmit your paper within 60 days. It will require further review. Please explain in your cover letter how you have changed the present version. If you require longer than 60 days to make the revisions, please contact Dr Cristina Ghiani (cghiani@mednet.ucla.edu). You can submit your revised manuscript directly by clicking on the following link: *** PLEASE NOTE: This is a two-step process. After clicking on the link, you will be directed to a webpage to confirm. ***

https://mc.manuscriptcentral.com/jnr?URL_MASK=8af5e8b986bc4d28ade61e8b1d0f9507

Thank you again for your submission to the Journal of Neuroscience Research; I look forward to reading your revised manuscript.

Best Wishes,

Dr Cristina Ghiani
CO-Editor-in-Chief, Journal of Neuroscience Research

Editorial Comments:
GRAPHICAL ABSTRACT

Please upload a graphical abstract, which we are asking of all authors submitting original research articles. This is intended to provide readers with a visual representation of the conclusions and an additional way to access the contents and appreciate the main message of the work. What we require is a .tif image file and a .doc text file containing an abbreviated abstract. For the image, labels, although useful, must be kept to a minimum and the image should be 400 x 300, 300 x 400, or 400 x 400 pixels square and at a resolution of 72 dpi. This can be one of the figures from your article, or something slightly different, as long as it represents your study. Instructions for this can be found in our author guidelines online at [http://onlinelibrary.wiley.com/journal/10.1002/\(ISSN\)1097-4547/homepage/ForAuthors.html](http://onlinelibrary.wiley.com/journal/10.1002/(ISSN)1097-4547/homepage/ForAuthors.html)

STATISTICAL TESTS

Articles containing statistical analyses should state the name of the statistical test, the n value for each statistical analysis, the comparisons of interest, and justification for the use of the test. It should be clear what statistical test was used to generate every P value. Moreover, the authors must include the values from the appropriate statistical test (e.g., $F(x,x) = xx$; $n = x$; $P = x.xxx$). If the tests violate any assumptions, the authors must provide this information.

This journal requests that p-values be shown using a consistent decimal exactness: values in this text are given to varying number of digits or they are expressed only as "P<0.05" and the like. Kindly choose a decimal, say 3, and stay with it throughout, with "p<0.001" reserved as appropriate.

DATA VISUALIZATION

JNR does not support the use of bar graphs, kindly change all these graphs to scatterplots or box and whisker plots rather than bar graphs to better visualize the distribution of data.

Please also be sure to review the following, to ensure accurate graphical visualization and transparent reporting: <https://goo.gl/w5dnYa> and <https://onlinelibrary.wiley.com/doi/epdf/10.1002/jnr.24340>

DATA ACCESSIBILITY

To enable readers to locate archived data from Journal of Neuroscience Research papers, we require authors to include a 'Data Accessibility' section just before the References. This should list the database(s) and URL(s) or dataset DOIs for all data associated with the manuscript. Data deposit repositories might include unstructured repositories such as Dryad, FigShare, NeuroMorpho or centralized repositories from the institutions in which the research was conducted. We also strongly recommend depositing data in the Open Science Framework. JNR will also allow small data sets to be included as Supplementary Files with the article.

Reviewer: 1

Comments to the Author

Comments on the manuscript:

This study investigates the effect of neurotrophins on OPCs cultured on CSPG (to model a glial scar). Although there is some literature available on the effect of neurotrophins on OPCs, the analysis of OPC morphology upon neurotrophin stimulation in the presence of CSPG, that has been shown to inhibit process outgrowth, is an addition to the current knowledge of regenerating processes after spinal cord injury.

The authors have shown in a previous study that CSPGs inhibit OPC outgrowth and differentiation.

- It is very interesting, that the addition of neurotrophins seems to stimulate differentiation of OPCs in the presence of CSPG. However, this effect was only assessed in the presence of CSPG. Nevertheless, the authors should provide the results of neurotrophin stimulation on CSPG in direct comparison with a neutral or growth-promoting substrate side by side. Only then will it be possible to show the extent by which neurotrophins reverse the inhibitory effect of CSPG relative to a substrate without CSPG or a permissive substrate, e.g. laminin (which is used as a control in Figure 1).

In addition, these experiments will add information to the fact that neurotrophins did not stimulate proliferation in the presence of CSPG. As the study is performed now, it is not entirely clear whether the applied neurotrophins would exert the same or more pronounced effects on OPCs, or even stimulate proliferation without being plated on CSPGs in the system that is used in this study.

- It is especially interesting that CNTF had no effect in the presence of CSPGs, as this factor has been previously reported to positively influence remyelination. It would be a good control to culture OPCs on laminin coated plates and add CNTF, to see whether the lack of response is really due to the presence of CSPG.

- The effect of neurotrophins on differentiation could be strengthened by including a marker for more mature oligodendrocyte lineage cells, e.g. APC/CC1.

- There is already literature published on the positive effect of neurotrophins on remyelination in several animal models. These studies should be mentioned in the introduction. Some information about previous literature is mentioned in the results part.

Could the authors comment on the discrepancies between these studies and the effect they are describing: which mechanisms/pathways do they think are blocked by the presence of CSPGs? How do neurotrophins override the negative effects of CSPGs?

- The figure legends are very long and repeat the messages written in the results part. They could be shortened and reduced to technical information.

- Page 8 line 36: what is meant with differentiation medium? The same as on page 7 line 3?

were the neurotrophins added to this differentiation medium? Is the defined medium on page 16 line 8 the same?

- Some redundancy in the discussion about CNTF

- Last sentence of discussion (page 20 line 20-25) should be better explained

Methods:

- The authors state that they did the experiments in duplicate. It is not clear how many replicates/coverlips were plated per condition, e.g. whether each experiment was performed in triplicate, or only one coverslip per condition. Could the authors please clarify this?

- The authors state for each experiment, that there was no effect on proliferation or cell death, based solely on the cell counts. It should be considered that there will be some microglia and astrocytes in the cultures generated by the method applied. Thus, if they want to make a clear statement about proliferation of OPCs, they must include a marker for oligodendrocyte lineage cells, e.g. Olig 2, and determine the proportion of true oligodendroglia in the experiments.

- In addition, the density of the cells in culture will affect their differentiation. If the cells are plated

densely, they will differentiate more. The density of the cultures has to be taken into account when assessing the effect of neurotrophins on survival, proliferation and process extension.

Reviewer: 2

Comments to the Author

In this manuscript, the authors use an established in vitro glial scar assay to explore the effects of neurotrophic growth factors on the differentiation of oligodendrocyte progenitors in the presence of inhibitory signals from proteoglycans (CSPG) to approximate a glial scar. This is a very simple and clean study and the objectives are clear. The knowledge obtained is important from the perspective of translational medicine as clinical treatments are being developed for patients and the data presented are of excellent quality. The experiments are internally consistent with small and consistent error bars and the figures are well presented and intuitive.. The authors should address a minor issue with the reporting of statistics.

1) the statistics in this study involve 2 way ANOVA, but F statistics are not reported. These should be included (including degrees-of-freedom), for example in the figure legends, to clearly define the analysis designs for each experiment.

*****IMPORTANT: Instructions and checklists follow*****

When finalized, please upload your complete revised manuscript onto our website, preferably as a word document. Please ensure to upload a highlighted version of your manuscript along with the clean version. The highlighted version should highlight the revised text or any other changes made to the manuscript. The clean version should have no highlighted sentences, strike-through words, or comments in margins. Kindly avoid submitting a document with tracked changes. Figures must be uploaded separately in .tif or eps format. Please review our submission checklist, which can be found in our author guidelines and also be sure to fill out the Transparent Science Questionnaire attached to this email.

To assist in our transparent peer review process, please upload the decision letter and response to reviewers using the attached template document. The first section should include the decision letter in full. In the Authors Response section, please provide the reviewer comments and your response in red.

JNR offers Open Science badges to qualifying authors. For more information please see the “Open Science initiatives” section of our author guidelines. If you would like to apply for one or more of the badges, please complete the included disclosure form and upload it as Supplemental Material Not for Review when submitting your final manuscript files.

To submit your revised manuscript:

*You have two ways to submit your revision. Either click on the link: *** PLEASE NOTE: This is a two-step process. After clicking on the link, you will be directed to a webpage to confirm. ***

https://mc.manuscriptcentral.com/jnr?URL_MASK=41ee720d9f4d42de9a598445dd7a6700 or Log on to <https://mc.manuscriptcentral.com/jnr> using your case-sensitive User ID ((Person not available)).

For security purposes your password is not listed in this email. If you are unsure of your password you

may click the link to set a new password.
(Person not available)

*Enter the Author Center, where you will find your manuscript title listed under “Manuscripts with Decisions”

* Under “Actions,” click on “Create a Revision”. Your manuscript number has been appended to denote a revision.

*Follow the prompts and replace existing files with revised ones, as necessary.

If you encounter any troubles in submitting your revised manuscript, please contact Dr Cristina Ghiani (cghiani@mednet.ucla.edu) or click on Get Help Now at the top right of any ScholarOne Manuscripts screen.

Author’s response

2nd Editorial Decision Minor revision

18-Oct-2020

Dear Dr Siebert:

Thank you for re-submitting your manuscript to the Journal of Neuroscience Research. We've now received the reviewers' feedback and have appended those reviews below. The reviewers are overall very enthusiastic and supportive of the study, however, there are still some concerns that need to be addressed by the authors along with additional changes. I expect that these points should be relatively straightforward to address. If there are any questions or points that are problematic, please feel free to contact me. I am glad to discuss.

We ask that you return your manuscript within 30 days. Please explain in your cover letter how you have changed the present version and a point by point response to the editor and reviewers' comments. If you require longer than 30 days to make the revisions, please contact Dr Cristina Ghiani (cghiani@mednet.ucla.edu). To submit your revised manuscript: Log in by clicking on the link below

(If the above link space is blank, it is because you submitted your original manuscript through our old submission site. Therefore, to return your revision, please go to our new submission site here (submission.wiley.com/jnr) and submit your revision as a new manuscript; answer yes to the question “Are you returning a revision for a manuscript originally submitted to our former submission site (ScholarOne Manuscripts)? If you indicate yes, please enter your original manuscript’s Manuscript ID number in the space below” and including your original submission's Manuscript ID number (jnr-2020-Apr-8757.R1) where indicated. This will help us to link your revision to your original submission.)

Thank you again for your submission to the Journal of Neuroscience Research; we look forward to reading your revised manuscript.

Best Wishes,

Dr Cristina Ghiani
Editor-in-Chief, Journal of Neuroscience Research

Editorial Comments to the Author:

1) The authors have not submitted a point by point response to the comments of the reviewers for the previous version to thoroughly explained how the points and concerns raised by the reviewers have been addressed. In the decision email it was clearly stated that "To assist in our transparent peer review process, please upload the decision letter and response to reviewers using the attached template document. The first section should include the decision letter in full. In the Authors Response section, please provide the reviewer comments and your response in red.

One the reviewers had requested some additional experiments, it appears that the authors have answered some of these comments by altering or adding text passages, but did not perform any additional experiments. Kindly explain and justify how these criticisms have been address and why the additional experiments were not performed.

Please submit a point by point response to the previous comments and to those enclosed below.

2) The following requests have not been addressed:

A. STATISTICAL TESTS

Articles containing statistical analyses should state the name of the statistical test, the n value for each statistical analysis, the comparisons of interest, and justification for the use of the test. It should be clear what statistical test was used to generate every P value. Moreover, the authors must include the values from the appropriate statistical test (e.g., $F(x,x) = xx$; $n = x$; $P = x.xxx$). If the tests violate any assumptions, the authors must provide this information.

This journal requests that p-values be shown using a consistent decimal exactness: values in this text are given to varying number of digits or they are expressed only as "P<0.05" and the like. Kindly choose a decimal, say 3, and stay with it throughout, with "p<0.001" reserved as appropriate.

B. DATA VISUALIZATION

JNR does not support the use of bar graphs, kindly change all these graphs to scatterplots or box and whisker plots rather than bar graphs to better visualize the distribution of data.

Please also be sure to review the following, to ensure accurate graphical visualization and transparent reporting:

<https://nam10.safelinks.protection.outlook.com/?url=https%3A%2F%2Fgoo.gl%2Fw5dnYa&data=04%7C01%7Cjustin.siebert%40sru.edu%7C37900a4ff90642a21fab08d87320ff37%7C86555dba073b4ff7b7d1b73a77c5bd92%7C0%7C0%7C637385934046214666%7CUnknown%7CTWFpbGZsb3d8eyJWljojMC4wLjAwMDAiLCJQIjoiV2luMzliLCJBTiI6Ikl1haWwiLCJXVCi6Mn0%3D%7C1000&reserved=0> and <https://nam10.safelinks.protection.outlook.com/?url=https%3A%2F%2Fonlinelibrary.wiley.com%2Fdoi%2F10.1002%2Fjnr.24340&data=04%7C01%7Cjustin.siebert%40sru.edu%7C37900a4ff90642a21fab08d87320ff37%7C86555dba073b4ff7b7d1b73a77c5bd92%7C0%7C0%7C637385934046214666%7CUnknown%7CTWFpbGZsb3d8eyJWljojMC4wLjAwMDAiLCJQIjoiV2luMzliLCJBTiI6Ikl1haWwiLCJXVCi6Mn0%3D%7C1000&reserved=0>

[%3D%7C1000∓sdata=a20tgTjyL8UckCip%2BDY%2BCoc99HV0RxcvN4O6vRK6Wos%3D&reserved=0](https://nam10.safelinks.protection.outlook.com/?url=http%3A%2F%2Fonlinelibrary.wiley.com%2Fjournal%2F10.1002%2F&data=04%7C01%7Cjustin.siebert%40sru.edu%7C37900a4ff90642a21fab08d87320ff37%7C86555dba073b4ff7b7d1b73a77c5bd92%7C0%7C0%7C637385934046224624%7CUnknown%7CTWFpbGZsb3d8eyJWljoiMC4wLjAwMDAiLCJQIjoiV2luMzliLCJBTiI6k1haWwiLCJXVCi6Mn0%3D%7C1000∓sdata=PZCHWV9nBQQ9na9zRMaKu%2Fiu54PCRYpTFRNqogIzsf8%3D&reserved=0(ISSN)1097-4547/homepage/ForAuthors.html)

C. GRAPHICAL ABSTRACT

Please upload a graphical abstract, which we are asking of all authors submitting original research articles. This is intended to provide readers with a visual representation of the conclusions and an additional way to access the contents and appreciate the main message of the work. What we require is a .tif image file and a .doc text file containing an abbreviated abstract. For the image, labels, although useful, must be kept to a minimum and the image should be 400 x 300, 300 x 400, or 400 x 400 pixels square and at a resolution of 72 dpi. This can be one of the figures from your article, or something slightly different, as long as it represents your study. Instructions for this can be found in our author guidelines online at [https://nam10.safelinks.protection.outlook.com/?url=http%3A%2F%2Fonlinelibrary.wiley.com%2Fjournal%2F10.1002%2F&data=04%7C01%7Cjustin.siebert%40sru.edu%7C37900a4ff90642a21fab08d87320ff37%7C86555dba073b4ff7b7d1b73a77c5bd92%7C0%7C0%7C637385934046224624%7CUnknown%7CTWFpbGZsb3d8eyJWljoiMC4wLjAwMDAiLCJQIjoiV2luMzliLCJBTiI6k1haWwiLCJXVCi6Mn0%3D%7C1000∓sdata=PZCHWV9nBQQ9na9zRMaKu%2Fiu54PCRYpTFRNqogIzsf8%3D&reserved=0\(ISSN\)1097-4547/homepage/ForAuthors.html](https://nam10.safelinks.protection.outlook.com/?url=http%3A%2F%2Fonlinelibrary.wiley.com%2Fjournal%2F10.1002%2F&data=04%7C01%7Cjustin.siebert%40sru.edu%7C37900a4ff90642a21fab08d87320ff37%7C86555dba073b4ff7b7d1b73a77c5bd92%7C0%7C0%7C637385934046224624%7CUnknown%7CTWFpbGZsb3d8eyJWljoiMC4wLjAwMDAiLCJQIjoiV2luMzliLCJBTiI6k1haWwiLCJXVCi6Mn0%3D%7C1000∓sdata=PZCHWV9nBQQ9na9zRMaKu%2Fiu54PCRYpTFRNqogIzsf8%3D&reserved=0(ISSN)1097-4547/homepage/ForAuthors.html)

Reviewer: 2

Comments to the Author
none

Reviewer: 1

Comments to the Author

This study investigates the effect of neurotrophins on OPCs cultured on CSPG (to model a glial scar). Although there is some literature available on the effect of neurotrophins on OPCs, the analysis of OPC morphology upon neurotrophin stimulation in the presence of CSPG, that has been shown to inhibit process outgrowth, is an addition to the current knowledge of regenerating processes after spinal cord injury.

The authors have shown in a previous study that CSPGs inhibit OPC outgrowth and differentiation. The presented model is suitable to analyze the effect of neurotrophins (and possibly also other factors) on OPC behavior. As it is presented right now, the manuscript remains purely descriptive, based on morphology, which is a first indicator to the impact of the neurotrophins tested on OPCs. The manuscript slightly improved compared to the first version.

There are still some comments on the manuscript.

- To check the purity of the OPC preparation, it is ok to use A2B5 or PDGFR α antibodies. But one cannot claim there are no microglia, if microglia-specific antibodies have not been tested. Most authors report some microglial contamination in their OPC cultures.
- The same is true for the differentiation. The authors did not assess the differentiation status by applying stage specific markers, but they judge this by morphology. So you are just describing process outgrowth and can only speculate about the differentiation status.
- Regarding the cell numbers that are assessed for every treatment, it is very vague to make a statement about proliferation and cell death simply based on cell numbers. Again, there were no markers checked for each of these processes.

- Laminin and Laminin/CSPG as a substrate have only been directly compared without the addition of neurotrophins in figure 1. The effects of BDNF, GDNF and NT-3 have only been tested on Laminin/CSPG. For CNTF, the authors write that they tested it on Laminin only although they do not show the data (it would be an addition to the study to show those data). As CNTF has no or limited effect on the morphology of OPCs in their hands, it would have been even more important to show those factors that had an effect on OPCs morphology on Laminin and Laminin/CSPG.
- It is ok to compare different in vitro studies in your discussion. But I think that especially in in vitro studies one has to carefully compare all control and treatment conditions, due to their high variability in different system.
- Check your reference list, newly inserted citations are not all listed
- The figure legends are still too long. In Figure1, the difference between I and J is not clear from the graph.
- OPCs in vivo are not bipolar. This must be a feature of the culture system you use.

*****IMPORTANT: Instructions and checklists follow***** When finalized, please upload your complete revised manuscript onto our website, preferably as a word document. Please ensure to upload a highlighted version of your manuscript along with the clean version. The highlighted version should highlight the revised text or any other changes made to the manuscript. The clean version should have no highlighted sentences, strike-through words, or comments in margins. Kindly avoid submitting a document with tracked changes. Figures must be uploaded separately in .tif or eps format. Please review our submission checklist, which can be found in our author guidelines and also be sure to fill out the Transparent Science Questionnaire attached to this email.

JNR offers Open Science badges to qualifying authors. For more information please see the “Open Science initiatives” section of our author guidelines. If you would like to apply for one or more of the badges, please complete the included disclosure form and upload it as Supplemental Material Not for Review when submitting your final manuscript files.

If you encounter any troubles in submitting your revised manuscript, please contact Dr Cristina Ghiani (cghiani@mednet.ucla.edu).

Authors' Response
11/18/2020

Editorial Comments to the Author:

1) The authors have not submitted a point by point response to the comments of the reviewers for the previous version to thoroughly explained how the points and concerns raised by the reviewers have been addressed. In the decision email it was clearly stated that "To assist in our transparent peer review process, please upload the decision letter and response to reviewers using the attached template document. The first section should include the decision letter in full. In the Authors Response section, please provide the reviewer comments and your response in red.

My co-author and I are puzzled by this comment as during our resubmission of the manuscript, we thought we had uploaded our point-by-point response. We apologize for this error.

One the reviewers had requested some additional experiments, it appears that the authors have answered some of these comments by altering or adding text passages, but did not perform any additional experiments. Kindly explain and justify how these criticisms have been address and why the additional experiments were not performed.

Please submit a point by point response to the previous comments and to those enclosed below.

2) The following requests have not been addressed:

A. STATISTICAL TESTS

Articles containing statistical analyses should state the name of the statistical test, the n value for each statistical analysis, the comparisons of interest, and justification for the use of the test. It should be clear what statistical test was used to generate every P value. Moreover, the authors must include the values from the appropriate statistical test (e.g., $F(x,x) = xx$; $n = x$; $P = x.xxx$). If the tests violate any assumptions, the authors must provide this information.

After an e-mail conversation with the editor and chief of this journal (10/21/2020), it was determined that this comment was addressed, and required no further action.

This journal requests that p-values be shown using a consistent decimal exactness: values in this text are given to varying number of digits or they are expressed only as "P<0.05" and the like. Kindly choose a decimal, say 3, and stay with it throughout, with "p<0.001" reserved as appropriate.

After an e-mail conversation with the editor and chief of this journal (10/21/2020), it was determined that this comment was addressed, and required no further action.

B. DATA VISUALIZATION

JNR does not support the use of bar graphs, kindly change all these graphs to scatterplots or box and whisker plots rather than bar graphs to better visualize the distribution of data.

Please also be sure to review the following, to ensure accurate graphical visualization and transparent reporting:

<https://nam10.safelinks.protection.outlook.com/?url=https%3A%2F%2Fgoo.gl%2Fw5dnYa&data=04%7C01%7Cjustin.siebert%40sru.edu%7C37900a4ff90642a21fab08d87320ff37%7C86555dba073b4ff7b7d1b73a77c5bd92%7C0%7C0%7C637385934046214666%7CUnknown%7CTWFpbGZsb3d8eyJWljoic4wLjAwMDAiLCJQIjoiV2luMzliLCJBTiI6I1haWwiLCJXVCI6Mn0%3D%7C1000&sd=0&reserved=0> and
<https://nam10.safelinks.protection.outlook.com/?url=https%3A%2F%2Fonlinelibrary.wiley.com%2Fdoi%2F10.1002%2Fjnr.24340&data=04%7C01%7Cjustin.siebert%40sru.edu%7C37900a4ff90642a21fab08d87320ff37%7C86555dba073b4ff7b7d1b73a77c5bd92%7C0%7C0%7C637385934046224624%7CUnknown%7CTWFpbGZsb3d8eyJWljoic4wLjAwMDAiLCJQIjoiV2luMzliLCJBTiI6I1haWwiLCJXVCI6Mn0%3D%7C1000&sd=0&reserved=0>

Per the guidelines for authors: "Bar graphs are encourage only when presenting categorical or count data" The data presented in this study is just that, it is count data, therefore the utilization of bar graphs is the best way to visually represent our experimental data. After an e-mail conversation with the editor and chief of this journal (10/28/2020), the manuscript has been amended to include a series of data tables that include the detailed information indicating number of cell counted, total number of cells, and the standard deviation for each of the treatment conditions. These data tables have been incorporated into the body of the manuscript.

C. GRAPHICAL ABSTRACT

Please upload a graphical abstract, which we are asking of all authors submitting original research articles. This is intended to provide readers with a visual representation of the conclusions and an additional way to access the contents and appreciate the main message of the work. What we require is a .tif image file and a .doc text file containing an abbreviated abstract. For the image, labels, although useful, must be kept to a minimum and the image should be 400 x 300, 300 x 400, or 400 x 400 pixels square and at a resolution of 72 dpi. This can be one of the figures from your article, or something slightly different, as long as it represents your study. Instructions for this can be found in our author guidelines online at [https://nam10.safelinks.protection.outlook.com/?url=http%3A%2F%2Fonlinelibrary.wiley.com%2Fjournal%2F10.1002%2F&data=04%7C01%7Cjustin.siebert%40sru.edu%7C37900a4ff90642a21fab08d87320ff37%7C86555dba073b4ff7b7d1b73a77c5bd92%7C0%7C0%7C637385934046224624%7CUnknown%7CTWFpbGZsb3d8eyJWljoIMC4wLjAwMDAiLCJQIjoiV2luMzliLCJBTiI6I6k1haWwiLCJXVCi6Mn0%3D%7C1000&sdata=PZCHWV9nBQQ9na9zRMaKu%2Fiu54PCrypTFRNqoqlZsF8%3D&reserved=0\(ISSN\)1097-4547/homepage/ForAuthors.html](https://nam10.safelinks.protection.outlook.com/?url=http%3A%2F%2Fonlinelibrary.wiley.com%2Fjournal%2F10.1002%2F&data=04%7C01%7Cjustin.siebert%40sru.edu%7C37900a4ff90642a21fab08d87320ff37%7C86555dba073b4ff7b7d1b73a77c5bd92%7C0%7C0%7C637385934046224624%7CUnknown%7CTWFpbGZsb3d8eyJWljoIMC4wLjAwMDAiLCJQIjoiV2luMzliLCJBTiI6I6k1haWwiLCJXVCi6Mn0%3D%7C1000&sdata=PZCHWV9nBQQ9na9zRMaKu%2Fiu54PCrypTFRNqoqlZsF8%3D&reserved=0(ISSN)1097-4547/homepage/ForAuthors.html)

A graphical abstract has been created for this manuscript, and submitted

Reviewer: 2

Comments to the Author

None

We thank the reviewer for their time reviewing our manuscript and their comments that helped to improve the overall quality our manuscript.

Reviewer: 1

Comments to the Author

This study investigates the effect of neurotrophins on OPCs cultured on CSPG (to model a glial scar). Although there is some literature available on the effect of neurotrophins on OPCs, the analysis of OPC morphology upon neurotrophin stimulation in the presence of CSPG, that has been shown to inhibit process outgrowth, is an addition to the current knowledge of regenerating processes after spinal cord injury.

The authors have shown in a previous study that CSPGs inhibit OPC outgrowth and differentiation. The presented model is suitable to analyze the effect of neurotrophins (and possibly also other factors) on OPC behavior. As it is presented right now, the manuscript remains purely descriptive, based on morphology, which is a first indicator to the impact of the neurotrophins tested on OPCs. The manuscript slightly improved compared to the first version.

There are still some comments on the manuscript.

- To check the purity of the OPC preparation, it is ok to use A2B5 or PDGFRa antibodies. But one cannot claim there are no microglia, if microglia-specific antibodies have not been tested. Most authors report some microglial contamination in their OPC cultures.

Microglial contamination is routinely seen in primary cultures of OPCs. Our methods of OPC purification eliminate a large majority of the Microglial contamination. Furthermore, microglia can be easily identified in culture by their morphological characteristics, and we know they do not survive in our culture conditions of the experimental model. They die within 24-36 hours

- The same is true for the differentiation. The authors did not assess the differentiation status by applying stage specific markers, but they judge this by morphology. So you are just describing process outgrowth and can only speculate about the differentiation status.

Both the title of this manuscript, and in the body of the manuscript, we have mentioned repeatedly that this study was designed to examine the effect of NT treatment on OPC process outgrowth in the environment of a glial scar. It is well known in the oligodendroglial field that OPC differentiation is first marked by a significant increase in the extent and complexity of process outgrowth. This is accompanied by the production of myelin proteins, which appears to be regulated independent of process outgrowth (Osterhout et al 1999; Thomason et al 2020). The initial step in formation of the myelin sheath is contact of the process with the axon. Thus, process outgrowth is an important step in OPC differentiation and in fact, does not begin until you send a signal to the OPCs to begin differentiation. Based on the extensive body of work that describes these steps of OPC differentiation and myelination, we respectfully disagree with the reviewers comment. The introduction has been slightly edited to make sure it is clear that this study was focused on OPC process outgrowth and not the production of myelin proteins during OPC differentiation. We apologize for any confusion regarding this matter.

- Regarding the cell numbers that are assessed for every treatment, it is very vague to make a statement about proliferation and cell death simply based on cell numbers. Again, there were no markers checked for each of these processes.

- Laminin and Laminin/CSPG as a substrate have only been directly compared without the addition of neurotrophins in figure 1. The effects of BDNF, GDNF and NT-3 have only been tested on Laminin/CSPG. For CNTF, the authors write that they tested it on Laminin only although they do not show the data (it would be an addition to the study to show those data). As CNTF has no or limited effect on the morphology of OPCs in their hands, it would have been even more important to show those factors that had an effect on OPCs morphology on Laminin and Laminin/CSPG.

The control experiments mentioned in this critique were conducted as part of the experimental design. We did not see any significant effects on OPC differentiation when NTs were added to the cultures. We were a little surprised by this, as we expected to see an additive effect of NT3 and possibly CNTF. However, as we have stated, we are using laminin as a cell substrate for all experiments, and laminin itself stimulates OPC differentiation.

As requested by the reviewer, we summarized the findings from them control experiments in a paragraph that was added to the "CSPGs Significantly Inhibit OPC Process Outgrowth" results section of the manuscript. We did not make a separate figure as the more interesting data of this manuscript was the ability of specific growth factors to promote OPC process outgrowth when OPCs are in the presence of inhibitory substrates (specifically the CSPGs found in the site of a spinal cord lesion). The study of OPC process outgrowth on a variety of different growth surfaces has been done (Buttery and ffrench-Constant 1999), and combinations of substrates and NTs would be the topic for another study all together.

- It is ok to compare different in vitro studies in your discussion. But I think that especially in in vitro studies one has to carefully compare all control and treatment conditions, due to their high variability in different system.

We do agree with the reviewer about this; many investigators use slightly different culture media, treatment conditions, and even OPC sources eg mouse and rat, which do show different growth characteristics. However, we would also point out that these studies are often done with limited cell substrates, eg polylysine and laminin. Polylysine is most often used because it is considered 'inert' and laminin is known to enhance the differentiation of OPC in culture, regardless of the media.

- Check your reference list, newly inserted citations are not all listed
We thank the reviewer for bringing this to our attention. Our mistake, and the missing references have been added to our reference list.
- The figure legends are still too long. In Figure1, the difference between I and J is not clear from the graph.
We have again edited the figure legends for this paper, and hope that they are now a reasonable length to satisfy this reviewer. The figure legend for figure 1 has been edited for clarity between Figure 1I and Figure 1J.
- OPCs in vivo are not bipolar. This must be a feature of the culture system you use.

We respectfully disagree with the reviewer. If you examine numerous in vitro studies of OPCs isolated from neonatal rat or mouse cortex, you will see a lot of bipolar cells, as they are proliferating and migration in the first postnatal days. You will also see a range of morphologies because there are a small population that are started differentiating. You rarely see fully differentiated cells because of the age of the brain and the fact that the isolation technique destroys those cells.

Perhaps the reviewer is more familiar with the NG2 cells seen in the brain, which are more complex in morphology. The reviewer needs to take a look at Siebert et al Exp Neurology 2011, as in this study we do show the presence of bipolar OPCs in vivo. They do exist, and are numerous after injury. Check the cover.

3rd Editorial Decision

Dear Dr Siebert:

Thank you for resubmitting your manuscript to the Journal of Neuroscience Research. I'm glad to inform you that your manuscript has been accepted pending few minor changes. Please see below.

We ask that you return your manuscript within 15 days. Please explain in your cover letter how you have changed the present version. If you require longer than 15 days to make the revisions, please contact Dr Cristina Ghiani (cghiani@mednet.ucla.edu). To submit your revised manuscript: Log in by clicking on the link below <https://rex-prod.resxchange.com/submissionBoard/1/8a233742-9d3a-4adb-801a-27ee6937a445/current>

If the above link space is blank, it is because you submitted your original manuscript through our old submission site. Therefore, to return your revision, please go to our new submission site here [submission.wiley.com/jnr](https://www.submission.wiley.com/jnr) and submit your revision as a new manuscript; answer yes to the question "Are you returning a revision for a manuscript originally submitted to our former submission site (ScholarOne Manuscripts)? If you indicate yes, please enter your original manuscript's Manuscript ID number in the space below" and including your original submission's Manuscript ID number (jnr-2020-Nov-9222) where indicated. This will help us to link your revision to your original submission.)

Thank you again for your submission to the Journal of Neuroscience Research; we look forward to reading your revised manuscript.

Best Wishes,

Dr Cristina Ghiani
Editor-in-Chief, Journal of Neuroscience Research

Associate Editor: Ghiani, Cristina

Comments to the Author:

Please correct the following sentence in the methods by adding the correct ANOVA test used:

" Statistical analysis was performed using Microsoft Excel running an ANOVA to test for significant differences in our treatment groups, followed by a Tukey's post-hoc test, to determine the significance between individual treatments. Significance was identified to at least the $P = 0.05$ level."

For the two way ANOVA please clearly state the variables.

Figure 1, what statistical test was used?

Please add the statistical test used in the figure legends.

*****IMPORTANT: Instructions and checklists follow*****

When finalized, please upload your complete revised manuscript onto our website, preferably as a word document. Please ensure to upload a highlighted version of your manuscript along with the clean version. The highlighted version should highlight the revised text or any other changes made to the manuscript. The clean version should have no highlighted sentences, strike-through words, or comments in margins. Kindly avoid submitting a document with tracked changes. Figures must be uploaded separately in .tif or eps format. Please review our submission checklist, which can be found in our author guidelines and also be sure to fill out the Transparent Science Questionnaire attached to this email.

JNR offers Open Science badges to qualifying authors. For more information please see the "Open Science initiatives" section of our author guidelines. If you would like to apply for one or more of the badges, please complete the included disclosure form and upload it as Supplemental Material Not for Review when submitting your final manuscript files.

Author Response

November 18, 2020

Dr. Christina Ghiani

Editor-in-Chief

Journal of Neuroscience Research

Dear Dr. Ghiani

On behalf of my co-author, and myself we would like to resubmit the attached manuscript, entitled "Select Neurotrophins Promote Oligodendrocyte Progenitor Cell Process Outgrowth in the Presence of Chondroitin Sulfate Proteoglycans" for publication in the Journal of Neuroscience Research. The manuscript has been edited taking into consideration the most recent set comments made by the editorial board and both reviewers. We feel that the latest round of critiques has continued to improve the overall quality of this manuscript, and thank everyone for their time and critiques.

This latest version of the manuscript contains the following alterations, to address the comments provided by the editorial board and reviewers:

- Text edits to the introduction for clarity
- References in the body of the text to incorporate the addition of the new data tables
- Text edits to the Results section “CSPGs Significantly Inhibit OPC Process Outgrowth” to summarize the results of the requested control experiments by reviewer 1
- Correction of the reference list, adding the missing references
- Correction to the legend for figure 1 to clear up the confusion indicated by reviewer 1
- The creation of a graphical abstract

In depth, explanations and point-by-point rebuttals can be found in the response to authors.

Both my coauthor and myself have edited and revised this manuscript and approve of the newest version for publication, and we have no competing interests. It has not been submitted for publication in any other journal. We would like to publish this work in the Journal of Neuroscience Research, as its publishing format is well suited for our work.

Thank you for your assistance with the preparation of this manuscript and for its reconsideration.

Sincerely,

Justin R. Siebert, Ph.D., MS.M.Ed

4th Editorial Decision

Dear Dr Siebert:

Thank you for submitting your manuscript "Select Neurotrophins Promote Oligodendrocyte Progenitor Cell Process Outgrowth in the Presence of Chondroitin Sulfate Proteoglycans" by Siebert, Justin R.; Osterhout, Donna J..

You will be pleased to know that your manuscript has been accepted for publication without change. Thank you for submitting this excellent work to our journal.

In the coming weeks, the Production Department will contact you regarding a copyright transfer agreement and they will then send an electronic proof file of your article to you for your review and approval.

Please note that your article cannot be published until the publisher has received the appropriate signed license agreement. Within the next few days, the corresponding author will receive an email from Wiley's Author Services asking them to log in. There, they will be presented with the appropriate license for completion. Additional information can be found at <https://authorservices.wiley.com/author-resources/Journal-Authors/licensing-open-access/index.html>

Would you be interested in publishing your proven experimental method as a detailed step-by-step protocol? Current Protocols in Neuroscience welcomes proposals from prospective authors to disseminate their experimental methodology in the rapidly evolving field of neuroscience. Please submit your proposal here: <https://currentprotocols.onlinelibrary.wiley.com/hub/submitproposal>

Congratulations on your results, and thank you for choosing the Journal of Neuroscience Research for publishing your work. I hope you will consider us for the publication of your future manuscripts.

Sincerely,

Dr Cristina Ghiani
Associate Editor, Journal of Neuroscience Research

Dr Cristina Ghiani
Editor-in-Chief, Journal of Neuroscience Research